# OpenReview forum: "USB: A Comprehensive and Unified Safety Evaluation Benchmark for Multimodal Large Language Models"
_ICLR.cc/2026/Conference — ICLR 2026 Conference Withdrawn Submission_

### Official Review · Reviewer_ZMah · 2025-10-30

**Soundness:** 2
**Presentation:** 2
**Contribution:** 2
**Rating:** 2
**Confidence:** 3

**Summary:**

This paper introduces the Unified Safety Benchmark (USB), a new comprehensive benchmark designed to address the fragmented, limited, and inconsistent nature of existing safety evaluations for Multimodal Large Language Models (MLLMs). The authors construct a highly granular, three-tiered safety taxonomy of 61 risk categories and systematically evaluate MLLM performance across four distinct image-text modality combinations. To achieve this comprehensive coverage, they develop a sophisticated data synthesis pipeline to fill the gaps identified in prior benchmarks. A key contribution of USB is its dual-pronged evaluation, which jointly assesses a model's vulnerability to generating harmful content and its tendency to over-refuse benign queries, thus providing a more balanced view of safety alignment. The paper validates this new benchmark through an extensive evaluation of 17 leading open-source and commercial MLLMs, offering significant insights into their current safety limitations.

**Strengths:**

- The paper's primary strength is its sheer scale and meticulous organization. The creation of a 61-category risk taxonomy and the systematic analysis across four modality combinations provide an unprecedentedly fine-grained tool for diagnosing MLLM safety weaknesses. This represents an engineering effort of high quality.

- The authors don't just create a new dataset; they first perform a rigorous analysis of over 13 existing benchmarks to identify coverage gaps. Their data synthesis pipeline is then purposefully designed to fill these gaps, particularly for cross-modal risks (RIST/SIST), which they correctly identify as a key weakness in prior work.

**Weaknesses:**

- The main weakness is the limited conceptual novelty. The paper follows the established paradigm of creating a static dataset of prompts and evaluating model responses. While the scale is impressive, it essentially represents a superset of existing efforts rather than a new way of thinking about evaluation. The field is arguably saturated with safety benchmarks (e.g., HarmBench, MMSafetyBench, VLSafe, VLSBench, etc.). This work feels like an escalation in a "benchmark arms race" rather than a paradigm shift towards more dynamic, adaptive, or reasoning-based evaluation methodologies that could offer more durable insights.

- The evaluation protocol heavily relies on GPT-4o as the final arbiter of safety. The authors themselves note in Appendix C that GPT-4o achieves only 82% agreement with human annotators on a subset of their data. This 18% disagreement rate is substantial and casts doubt on the reliability of the fine-grained results across 61 different categories. It is plausible that the judge's accuracy varies significantly across different risk types and modalities, potentially skewing the results and model rankings in non-obvious ways. This dependency on a single, proprietary, and imperfect model is a significant limitation for a benchmark aiming to be a unified standard.

- As with all static benchmarks, USB is susceptible to becoming stale. As it gains popularity, developers may inadvertently or intentionally train their models to pass its specific test cases, a phenomenon known as Goodhart's Law. While the USB-Hard subset attempts to mitigate this, it is still a static collection. The paper does not propose a framework for continuous evolution or adaptation, which limits its long-term value in a rapidly advancing field.

**Questions:**

- Beyond scale and comprehensiveness, what do you consider to be the primary conceptual takeaway from your work for future benchmark designers? In other words, what fundamental principle of evaluation does USB introduce that is distinct from simply being a larger and more organized collection of prior ideas?

- Regarding the LLM-as-Judge limitation: Given the 82% agreement with human annotators, have you analyzed whether this disagreement is uniformly distributed across the 61 risk categories and 4 modality combinations? Could it be that GPT-4o is significantly less reliable for certain nuanced categories (e.g., ethical safety vs. public safety)?

- The creation of such a comprehensive benchmark is a massive effort. How do you envision maintaining USB's relevance over time? Are there plans or a proposed framework for community-driven updates to add new scenarios and adversarial examples as MLLMs evolve and new vulnerabilities are discovered?

- In your data curation process, you define adequate coverage as having a minimum of 20 data samples. Could you elaborate on how this threshold was determined? Was it based on a statistical power analysis or is it a heuristic choice? Understanding the rationale would help in assessing the statistical robustness of the evaluations for each fine-grained category.

---

### Official Review · Reviewer_aaks · 2025-10-30

**Soundness:** 3
**Presentation:** 3
**Contribution:** 2
**Rating:** 4
**Confidence:** 4

**Summary:**

This paper proposes USB, a comprehensive benchmark for evaluating the safety of multimodal large language models across 61 risk categories and 4 modality combinations, covering both vulnerability and over-refusal (oversensitivity). Experiments show that current MLLMs still struggle to balance safety and utility, exhibiting trade-offs between avoiding vulnerabilities and excessive refusals, and remain particularly vulnerable to multimodal risky inputs.

**Strengths:**

1. The paper provides a thorough diagnosis of the key weaknesses in existing MLLM safety benchmarks—including insufficient data quality, limited risk coverage, and the neglect of modality combinations—and proposes targeted solutions such as automated data validation, expanded risk taxonomy, and the systematic design of four modality configurations. These are genuine and timely challenges that the work addresses in a structured and convincing manner.

2. The authors make a commendable effort to aggregate and analyze prior benchmarks and to define a broader taxonomy of risk categories.

3. The evaluation covers a large number of models and modalities, which provides some practical value for comparing current MLLMs.

**Weaknesses:**

1. The paper primarily integrates existing benchmarks and standard practices into a larger dataset. The methodology—combining prior datasets, generating synthetic examples, and measuring refusal/safety rates—follows well-known recipes and lacks a genuinely new insight into safety measurement or model behavior.

2. The contribution reads more like a benchmark report than a scientific study. There is little theoretical motivation or analysis beyond data aggregation, and the results mostly confirm known trends (commercial models safer; trade-off between safety and refusal).

3. The paper does not discuss why certain models fail under specific modality combinations, nor does it propose new evaluation metrics or diagnostic insights. The “unified” benchmark concept remains descriptive rather than analytical.

**Questions:**

A sensitivity analysis on GPT-4o’s evaluation stability (e.g., by varying prompts, temperature, or random seeds) would help determine whether USB’s reported consistency genuinely reflects benchmark robustness or merely GPT-4o’s internal bias.

---

### Official Review · Reviewer_DiVA · 2025-10-31

**Soundness:** 2
**Presentation:** 2
**Contribution:** 2
**Rating:** 2
**Confidence:** 3

**Summary:**

This paper introduces USB (Unified Safety Benchmark), a comprehensive safety evaluation benchmark for Multimodal Large Language Models (MLLMs) that covers 61 risk subcategories across 4 modality combinations (risky-image/risky-text, risky-image/safe-text, safe-image/risky-text, safe-image/safe-text), and pioneers the simultaneous assessment of both model vulnerability and over-refusal.

**Strengths:**

Category coverage is a strength of the paper. Users only need to test on the Unified Safety Benchmark to obtain a comprehensive and reliable safety assessment without combining multiple benchmarks.

**Weaknesses:**

1. The paper states that current benchmarks have limited data volume, citing less than 5K as an example, but notes that 5K data points are already quite substantial for a benchmark.

2. It is unclear if the synthetic prompts accurately reflect real-world scenarios, particularly with the synthetic images, and whether they are sufficient for safety evaluation. This aspect appears to be a potential limitation of the paper.

3. The evaluated models do not appear comprehensive, and some may be outdated. As a result, the conclusions might not extend to all current models. Some mainstream models are not included, such as Gemini-2.5-pro.

4. The paper does not seem to introduce new insights. The main observation is that models exhibit a high degree of homogeneity across most categories, suggesting their vulnerabilities may be systematic rather than isolated cases. However, the paper does not analyze the underlying reasons for this observation.

**Questions:**

1. I suggest the authors add more recent models, for example, those released before September-October.
2. Given the benchmark's characteristic of comprehensiveness, I suggest using it to uncover new insights, such as whether certain models have unique advantages/disadvantages and why.

---

### Official Review · Reviewer_gzxS · 2025-11-09

**Soundness:** 2
**Presentation:** 3
**Contribution:** 3
**Rating:** 4
**Confidence:** 3

**Summary:**

The paper introduces USB, a unified safety benchmark for multimodal LLMs covering 61 tertiary risk categories crossed with four modality combinations (RIRT/RIST/SIRT/SIST), plus an over-refusal slice. USB combines curated prior datasets with a sizable synthetic pipeline to fill sparsely covered areas, yielding 13,175 (Base) and 3,785 (Hard) samples. Evaluations on 12 open-source and 5 commercial MLLMs report lower safety rates on USB than on prior benchmarks, expose weaknesses on image-only and cross-modal risks, and visualize the safety–refusal trade-off.

**Strengths:**

1.Coverage & structure. A clear two-axis design (taxonomy × modality combos) with explicit attention to RIST/SIST, which most benchmarks miss.
2.Difficulty & discrimination. Lower average SR across models than prior sets; USB-Hard further separates systems while correlating with USB-Base (ρ≈0.98).
3.Trade-off view. Joint reporting of vulnerability (SR) and over-refusal (RR) is practical and often neglected. Metrics are defined and computed consistently.

**Weaknesses:**

1.A large share of items is synthetically generated, which can imprint generator-specific artifacts and cause domain shift from real photos/screenshots. This clouds external validity. I recommend: report metrics by source (public vs. synthetic) and add out-of-domain tests (real photos/screens vs. synthetic).
2. The evaluation relies on a single closed-source judge, which risks judge bias and hides sensitivity to hyperparameters. I suggest: add a multi-judge setup (mix of open- and closed-source) with small-scale human checks, and report inter-judge agreement.

**Questions:**

1.Recent studies suggest VLMs can be especially brittle on real-world artifacts (memes, natural photos/screens) and in multi-turn settings compared with synthetic, single-turn cases. Could you share your thoughts on adding a curated OOD slice (real photos/screenshots/memes) and a session-level multi-turn track, and—if feasible—reporting the resulting OOD gaps and rank stability?

2.Many recent safety benchmarks adopt multi-judge ensembles, report inter-judge agreement (e.g., Cohen’s κ), and include calibration/meta-evaluation; some also provide bootstrap CIs and rank stability across decoding/safety settings. Would you consider complementing your evaluation—currently based on a single closed-source judge and default settings—with agreement and calibration analyses, plus CIs and stability checks across alternative decoding/safety configurations?

---

### Note · Authors · 2026-01-06

**Comment:**

Dear Program Chairs,
We kindly request to withdraw our submission (Title: "USB: A Comprehensive and Unified Safety Evaluation Benchmark for Multimodal Large Language Models").
Due to a change in our research direction and future publication plans, we have decided to withdraw at this time. This decision is purely based on strategic considerations.
We appreciate the efforts of the reviewers and organizing committee.

**Withdrawal Confirmation:**

I have read and agree with the venue's withdrawal policy on behalf of myself and my co-authors.